# Simplified hypertension screening methods across 60 countries: An observational study

**Rodrigo M. Carrillo-Larco** [1,2,3]*, **Wilmer Cristobal Guzman-Vilca** [2,4,5], **Dinesh Neupane** [6]

1 Department of Epidemiology and Biostatistics, School of Public Health, Imperial College London, London, United Kingdom, 2 CRONICAS Centre of Excellence in Chronic Diseases, Universidad Peruana Cayetano Heredia, Lima, Peru, 3 Universidad Continental, Lima, Peru, 4 School of Medicine "Alberto Hurtado," Universidad Peruana Cayetano Heredia, Lima, Peru, 5 Sociedad Científica de Estudiantes de Medicina de Cayetano Heredia, Universidad Peruana Cayetano Heredia, Lima, Peru, 6 Department of International Health, Johns Hopkins Bloomberg School of Public Health, Johns Hopkins University, Baltimore, Maryland, United States of America

* r.carrillo-larco@imperial.ac.uk

**Data Availability Statement:** All data files are available from https://extranet.who.int/ncdsmicrodata/index.php/home. This needs a usernames, password and application, but data are

## Abstract

### Background

Simplified blood pressure (BP) screening approaches have been proposed. However, evidence is limited to a few countries and has not documented the cardiovascular risk amongst missed hypertension cases, limiting the uptake of these simplified approaches. We quantified the proportion of missed, over-diagnosed, and consistently identified hypertension cases and the 10-year cardiovascular risk in these groups.

### Methods and findings

We used 60 WHO STEPS surveys (cross-sectional and nationally representative; $n =$ 145,174) conducted in 60 countries in 6 world regions between 2004 and 2019. Nine simplified approaches were compared against the standard (average of the last 2 of 3 BP measurements). The 10-year cardiovascular risk was computed with the 2019 World Health Organization Cardiovascular Risk Charts. We used $t$ tests to compare the cardiovascular risk between the missed and over-diagnosed cases and the consistent hypertension cases. We used Poisson multilevel regressions to identify risk factors for missed cases (adjusted for age, sex, body mass index, and 10-year cardiovascular risk). Across all countries, compared to the standard approach, the simplified approach that missed the fewest cases was using the second BP reading if the first BP reading was 130–145/80–95 mm Hg (5.62%); using only the second BP reading missed 5.82%. The simplified approach with the smallest over-diagnosis proportion was using the second BP reading if the first BP measurement was ≥140/90 mm Hg (3.03%). In many countries, cardiovascular risk was not significantly different between the missed and consistent hypertension groups, yet the mean was slightly lower amongst missed cases. Cardiovascular risk was positively associated with missed hypertension depending on the simplified approach. The main limitation of the work is the cross-sectional design.

open access and released within days from the application.

**Funding:** RMC-L is supported by a Wellcome Trust International Training Fellowship (Wellcome Trust 214185/Z/18/Z). DN receives support from Resolve to Save Lives, an initiative of Vital Strategies, which is funded by Bloomberg Philanthropies, the Bill & Melinda Gates Foundation, and Gates Philanthropy Partners, which is funded with support from the Chan Zuckerberg Initiative. The funders had no role in study design, data collection and analysis, decision to publish, or preparation of the manuscript.

**Competing interests:** The authors have declared that no competing interests exist.

**Abbreviations:** BMI, body mass index; BP, blood pressure; LMICs, low- and middle-income countries; PR, prevalence ratio.

## Conclusions

Simplified BP screening approaches seem to have low misdiagnosis rates, and cardiovascular risk could be lower amongst missed cases than amongst consistent hypertension cases. Simplified BP screening approaches could be included in large screening programmes and busy clinics.

## Author summary

### Why was this study done?

- To measure blood pressure, one usually needs to take 3 measurements, waiting 3 minutes between measurements, and compute the average of the last 2 measurements.

- Because this standard protocol takes time and reduces the number of people who can be screened for hypertension, simplified screening approaches have been proposed (e.g., only taking 2 measurements), yet these simplified approaches have not been studied worldwide.

### What did the researchers do and find?

- We considered 9 simplified approaches and computed the number of cases that would be missed—and the number of cases that would be over-diagnosed—in comparison to the standard protocol, and described the cardiovascular risk profile in these groups.

- Two simplified approaches had the smallest misdiagnosis rates, though these rates differed between countries: using only the second blood pressure measurement (not the average of the last 3) and using the second blood pressure measurement if the first one was 130–145/80–95 mm Hg.

### What do these findings mean?

- Worldwide, simplified blood pressure screening approaches appear to be reliable for hypertension screening without missing many cases.

- Countries should identify the best simplified screening approach according to the local blood pressure distribution, hypertension epidemiology, and available resources for massive blood pressure screening programmes targeting the general population.

## Introduction

High blood pressure (BP) [1,2] is highly prevalent and a major risk factor for cardiovascular morbidity and mortality worldwide; it disproportionally affects low- and middle-income countries (LMICs) [3], where screening for hypertension remains suboptimal [4]. Even though there are effective interventions and treatments for hypertension [5,6], patients first need to be

diagnosed, which in the most comprehensive scenario requires multiple BP measurements on separate occasions [7–9], and in the most pragmatic approach requires 3 BP measurements, taking the average of the last 2 (standard approach) [10,11]. Three BP measurements may still be challenging in resource-constrained settings because of workforce shortage, low follow-up of patients, poor literacy, and affordability. Therefore, fewer BP measurements being needed to diagnose hypertension would help in screening large populations (e.g., May Measurement Month [12,13]) while saving time and resources in resource-constrained settings. Attempts have been made to find simplified BP screening approaches, such as only taking a second BP measurement if the first one was above a given threshold [14]. However, because these simplified approaches [14] have been tested in only 3 countries and BP may vary between countries [1], it is unknown whether there would be good agreement between the standard and the simplified approaches for BP screening in diverse populations. Furthermore, there may be concerns about the cardiovascular risk profile of hypertension cases missed by the simplified approaches, that is, whether the simplified approaches are missing people at high cardiovascular risk who would benefit from antihypertensive medication and cardiovascular disease prevention. To advance the literature on simplified BP screening approaches with data from multiple countries and to characterise the cardiovascular risk profile of the cases missed by the simplified approaches—with the aim of strengthening the recommendations for simplified BP screening approaches—we analysed national surveys in 60 countries. We aimed to answer the following research questions: What are the misdiagnosis and over-diagnosis rates for 9 simplified BP screening approaches? And what is the underlying cardiovascular risk profile for misdiagnosed, over-diagnosed, and consistently diagnosed cases?

## Methods

### Data sources and study population

We analysed WHO STEPS surveys [10,15]. These are population-based surveys conducted in nationally representative samples. These surveys follow a standard questionnaire and protocol including anthropometric, BP (S16 Table), and biomarker measurement [10]. If there were multiple surveys in a country, we used only the most recent one (i.e., only 1 survey per country was analysed).

We selected surveys with 3 BP measurements and with information on current smoking status, body mass index (BMI), waist circumference, fasting plasma glucose, and total cholesterol. We included adults aged 18–69 years and free of known hypertension (self-reported medical history and antihypertensive medication) [14]. In other words, we excluded people with hypertension because the simplified approaches for BP measurement are meant to be used in large screening programmes targeting the general population with undiagnosed hypertension.

None of the analyses presented in this paper were prespecified in a protocol. This study is reported as per the Strengthening the Reporting of Observational Studies in Epidemiology (STROBE) guideline (S1 Checklist).

### Variable definition

The standard approach for BP screening uses the mean of the second and third BP measurements and defines hypertension as systolic BP $\geq$ 140 mm Hg or diastolic BP $\geq$ 90 mm Hg. This definition was compared to 9 simplified BP screening approaches: (i) first BP measurement; (ii) second BP measurement; (iii) average of the first and second BP measurements; (iv) second BP measurement if the first systolic BP is $\geq$130 mm Hg or the first diastolic BP is $\geq$80 mm Hg (otherwise the first BP measurement is used); (v) second BP measurement if the first

systolic BP is ≥135 mm Hg or the first diastolic BP is ≥85 mm Hg (otherwise the first BP measurement is used); (vi) second BP measurement if the first systolic BP is ≥140 mm Hg or the first diastolic BP is ≥90 mm Hg (otherwise the first BP measurement is used); (vii) second BP measurement if the first systolic BP is 130–145 mm Hg or the first diastolic BP is 80–95 mm Hg (otherwise the first BP measurement is used); (viii) second BP measurement if the first systolic BP is 130–150 mm Hg or the first diastolic BP is 80–100 mm Hg (otherwise the first BP measurement is used); and (ix) second BP measurement if the first systolic BP is 130–155 mm Hg or the first diastolic BP is 80–105 mm Hg (otherwise the first BP measurement is used).

We compared each of the 9 simplified approaches with the standard to define 4 groups: (i) missed diagnosis: hypertension with the standard but non-hypertension with the simplified approach; (ii) over-diagnosis: non-hypertension with the standard but hypertension with the simplified approach; (iii) consistent hypertension: hypertension with both the standard and the simplified approach; and (iv) consistent non-hypertension: non-hypertension with both the standard and the simplified approach. In simple terms, misdiagnosis refers to people who would have been diagnosed with the standard approach but were not with the simplified approach; similarly, over-diagnosis refers to people who would not have been diagnosed with the standard approach but were with the simplified approach.

The proportion of missed hypertension cases was defined as the number of missed cases divided by the number of missed cases plus the number of consistent hypertension cases. The proportion of over-diagnosed cases was defined as the number of over-diagnosed cases divided by the number of over-diagnosed cases plus the number of consistent non-hypertension cases.

To characterise the cardiometabolic profile of the missed and over-diagnosed hypertension cases, we calculated the 10-year cardiovascular risk with the 2019 World Health Organization Cardiovascular Risk Charts [16]; we used the Stata package developed by the University of Cambridge Cardiovascular Epidemiology Unit [17]. We did not intend to make projections about cardiovascular risk; rather, we used 10-year cardiovascular risk as a summary measure to characterise overall cardiometabolic profile according to the groups of interest. Alternatively, we would have had to describe each cardiometabolic risk factor, making the results cumbersome. To compute 10-year predicted absolute cardiovascular risk, we used the same predictors as in the original 2019 WHO Cardiovascular Risk Charts except for diabetes; the original model included history of diabetes, whereas we included history and new diabetes cases (i.e., aware and unaware cases, the latter defined with fasting plasma glucose ≥ 126 mg/dl or 7 mmol/L). We included new diabetes cases in characterising cardiovascular risk profile in the missed and over-diagnosed populations because, in LMICs, many people with diabetes are unaware of their diagnosis [18]. We considered 10-year predicted cardiovascular risk as a continuous variable (from 0 to 1; not categorised as low/high risk).

## Statistical analysis

This is a descriptive analysis conducted with R (version 4.1.1). Analysis scripts are available as S1 Cleaning and S1 Analysis and S1 WHO CVD Risk. A $p$-value ≤ 0.05 was considered statistically significant. We did not use sampling weights in the analyses because we aimed to compare groups of simplified BP screening approaches rather than reporting prevalences representative at the country level [14].

First, we summarised the proportions (%) of missed and over-diagnosed cases at the global, regional, and country level (countries in each region are listed in S1 Table). Second, we summarised mean 10-year cardiovascular risk according to the 4 groups: missed diagnosis, over-diagnosis, consistent hypertension, and consistent non-hypertension. We used $t$ tests to compare the mean cardiovascular risk in the missed, over-diagnosed, and consistent non-hypertension

groups against that in the consistent hypertension group. The *p*-values for these *t* tests are reported for crude analyses and for analyses adjusted for multiple comparisons (Bonferroni). Third, to identify potential correlates of being misdiagnosed, we fitted individual-level Poisson multilevel regression models. The outcome was misdiagnosis according to each of the 9 simplified approaches (yes/no); thus, we had 9 regression models. The independent variables (i.e., predictors) were sex (reference was men), age (years), BMI (kg/m$^2$), and 10-year cardiovascular risk. Random intercepts were included whereby countries were nested within regions. The regression results are presented as prevalence ratios (PRs) along with 95% confidence intervals (95% CIs).

### Ethics

We analysed de-identified open-access data [15]. We did not request approval from an ethics committee. The authors alone are responsible for the opinions in the paper. The funder had no role in the study design, analysis, results interpretation, or conclusions. The first 2 authors conducted the analyses and vouch for the accuracy of the results.

## Results

### Data description

We analysed 60 STEPS surveys from 60 countries including 145,174 individuals (S1 Supplementary Flow Chart); the smallest sample size was 215 people in British Virgin Islands, and the largest was 7,431 people in Ethiopia (Table 1). The mean age ranged from 34 to 44 years, and the number of men ranged from 90 to 3,164. In all countries, the first BP record was higher than the average of the second and third records; the mean difference between the first and the average of the last 2 readings ranged from 0.70 mm Hg (Lebanon) to 5.75 mm Hg (Cambodia).

### Misdiagnosis

Across all countries, compared to the standard approach, the simplified approach that missed the fewest cases was using the second BP reading if the first BP measurement was 130–145/80–95 mm Hg; for this simplified approach, the misdiagnosed proportion was 5.62% (Table 2). The approach of using only the second BP reading missed 5.82% of cases. The other simplified approaches missed more than 6% of hypertension cases; the simplified approach that missed the most cases was using the second BP reading if the first BP measurement was ≥140/90 mm Hg (15.15% misdiagnosis).

  The same pattern was observed at the region level (Table 3). In 4 out of the 6 regions, the smallest proportion of misdiagnosis was found with the second BP record if the first BP measurement was 130–145/80–95 mm Hg: from 3.64% (Europe) to 9.25% (Americas). In the 2 remaining regions, the smallest proportion of misdiagnosis was found with the second BP reading only: 4.99% in the Eastern Mediterranean and 5.40% in Africa. Of note, across all regions the largest misdiagnosis proportion was found when using the second BP reading if the first BP measurement was ≥140/90 mm Hg.

  At the country level, the same pattern arose (Fig 1). In general, the smallest proportions of misdiagnosis were found with the second BP reading if the first BP measurement was 130–145/80–95 mm Hg or with the second BP measurement only. For both simplified approaches, Kuwait had the smallest proportion of misdiagnosis (1%), whilst Ecuador had the largest (14% and 12%, respectively). Notably, the misdiagnosis proportions were high when using the second BP reading if the first BP measurement was ≥140/90 mm Hg.

  Overall, based solely on misdiagnosis, using the second BP record if the first BP measurement is 130–145/80–95 mm Hg, or the second BP reading only, seem to be reasonable simplified

**Table 1. Descriptive statistics of the analysed surveys.**

| Country | Year | Sample | Men | Age (years) | First SBP (mm Hg) | Second SBP (mm Hg) | Third SBP (mm Hg) | BMI (kg/m²) | Waist circumference (cm) | FPG (mmol/L) | TC (mmol/L) | 10-year CV risk |
|---|---|---|---|---|---|---|---|---|---|---|---|---|
| Afghanistan | 2018 | 2,550 | 1,503 | 36 | 127.1 | 124.0 | 122.6 | 24.7 | 86.3 | 5.1 | 3.8 | 4.1 |
| Algeria | 2017 | 4,653 | 2,234 | 39 | 129.0 | 125.8 | 124.7 | 26.4 | 92.3 | 5.4 | 4.2 | 5.0 |
| American Samoa | 2004 | 1,137 | 503 | 42 | 134.2 | 131.5 | 130.6 | 35.3 | 105.3 | 7.2 | 4.8 | 5.2 |
| Armenia | 2016 | 1,205 | 355 | 42 | 131.0 | 129.3 | 127.9 | 26.5 | 90.9 | 4.6 | 4.4 | 4.9 |
| Azerbaijan | 2017 | 1,772 | 782 | 42 | 126.5 | 124.5 | 122.9 | 26.6 | 89.9 | 5.1 | 4.4 | 4.3 |
| Bangladesh | 2018 | 5,688 | 2,773 | 38 | 121.0 | 118.5 | 117.3 | 22.7 | 78.7 | 5.4 | 4.4 | 2.6 |
| Belarus | 2017 | 2,744 | 1,253 | 41 | 130.5 | 128.6 | 127.5 | 25.8 | 85.9 | 4.6 | 4.7 | 5.8 |
| Benin | 2015 | 4,397 | 2,116 | 37 | 130.3 | 127.1 | 125.0 | 23.5 | 81.9 | 4.9 | 4.0 | 2.0 |
| Bhutan | 2019 | 4,149 | 1,622 | 39 | 125.5 | 123.2 | 122.1 | 25.2 | 82.7 | 4.6 | 3.7 | 2.2 |
| Botswana | 2014 | 2,231 | 811 | 34 | 127.9 | 125.1 | 123.9 | 23.8 | 80.6 | 4.5 | 3.7 | 1.8 |
| British Virgin Islands | 2009 | 215 | 90 | 41 | 129.7 | 127.2 | 126.2 | 28.6 | 91.0 | 5.9 | 4.8 | NA |
| Brunei Darussalam | 2016 | 1,242 | 526 | 37 | 124.1 | 120.4 | 119.3 | 26.5 | 85.1 | 5.1 | 5.1 | 2.8 |
| Cabo Verde | 2007 | 693 | 278 | 42 | 138.5 | 134.5 | 132.6 | 24.6 | 85.3 | 5.6 | 4.2 | 2.8 |
| Cambodia | 2010 | 4,403 | 1,647 | 42 | 120.1 | 115.0 | 113.7 | 21.7 | 75.4 | 4.0 | 4.5 | 2.3 |
| Comoros | 2011 | 1,139 | 280 | 41 | 133.2 | 128.5 | 126.8 | 26.3 | 88.6 | 4.1 | 4.6 | 2.5 |
| Cook Islands | 2015 | 567 | 245 | 39 | 131.5 | 127.8 | 125.6 | 33.5 | 103.1 | 6.3 | 5.0 | NA |
| Ecuador | 2018 | 3,172 | 1,361 | 39 | 120.7 | 118.1 | 117.0 | 27.0 | 89.1 | 5.2 | 4.4 | 2.0 |
| Eritrea | 2010 | 4,152 | 1,103 | 43 | 121.2 | 117.8 | 116.0 | 20.2 | 75.5 | 4.1 | 4.8 | 2.5 |
| Eswatini | 2014 | 1,759 | 701 | 36 | 127.5 | 124.8 | 122.8 | 26.0 | 84.3 | 5.1 | 3.8 | 2.0 |
| Ethiopia | 2015 | 7,431 | 3,164 | 35 | 123.1 | 120.5 | 119.2 | 20.8 | 75.5 | 4.5 | 3.6 | 1.5 |
| Georgia | 2016 | 1,855 | 551 | 44 | 128.2 | 125.5 | 124.0 | 27.9 | 90.8 | 4.5 | 4.4 | 5.0 |
| Guyana | 2016 | 572 | 227 | 38 | 125.2 | 123.4 | 122.4 | 26.4 | 91.8 | 5.0 | 5.1 | 2.8 |
| Iraq | 2015 | 3,166 | 1,301 | 39 | 131.5 | 130.0 | 129.7 | 29.3 | 97.7 | 5.9 | 4.8 | 6.5 |
| Jordan | 2019 | 2,320 | 843 | 38 | 118.3 | 114.7 | 114.3 | 28.2 | 90.7 | 4.6 | 3.9 | 4.5 |
| Kenya | 2015 | 3,463 | 1,462 | 37 | 128.0 | 124.8 | 123.0 | 23.1 | 78.7 | 4.7 | 3.7 | 1.9 |
| Kiribati | 2016 | 966 | 429 | 38 | 130.1 | 126.0 | 125.0 | 29.3 | 90.8 | 6.0 | 4.0 | 4.0 |
| Kuwait | 2014 | 1,461 | 576 | 35 | 118.9 | 117.5 | 117.2 | 29.1 | 89.3 | 5.5 | 5.0 | 4.0 |
| Kyrgyzstan | 2013 | 1,815 | 721 | 41 | 132.8 | 130.0 | 128.2 | 26.6 | 87.9 | 4.8 | 4.4 | 3.7 |
| Lao People's Democratic Republic | 2013 | 2,156 | 880 | 38 | 119.6 | 116.5 | 115.6 | 22.7 | 77.2 | 4.5 | 4.2 | 2.1 |
| Lebanon | 2017 | 793 | 298 | 44 | 126.1 | 125.5 | 125.3 | 27.6 | 94.1 | 5.4 | 5.5 | 8.6 |
| Lesotho | 2012 | 1,493 | 552 | 41 | 130.6 | 127.2 | 125.6 | 25.5 | 83.1 | 4.5 | 3.5 | 2.5 |
| Libya | 2009 | 1,408 | 805 | 40 | 138.8 | 135.2 | 134.0 | 27.6 | 94.4 | 5.4 | 4.6 | 6.6 |
| Malawi | 2017 | 3,199 | 1,231 | 37 | 123.8 | 120.7 | 119.3 | 22.9 | 77.3 | 4.7 | 3.7 | 1.9 |
| Mongolia | 2019 | 4,019 | 1,899 | 39 | 121.8 | 119.3 | 118.1 | 25.7 | 86.7 | 5.8 | 4.4 | 3.3 |
| Morocco | 2017 | 3,540 | 1,282 | 41 | 130.8 | 127.8 | 126.3 | 26.4 | 93.1 | 5.6 | 3.6 | 5.4 |
| Myanmar | 2014 | 5,541 | 2,069 | 43 | 125.1 | 122.6 | 121.8 | 22.1 | 76.7 | 5.1 | 4.6 | 3.0 |
| Nauru | 2016 | 706 | 333 | 35 | 123.4 | 120.2 | 120.2 | 34.2 | 102.4 | 5.4 | 3.9 | NA |
| Nepal | 2019 | 4,428 | 1,546 | 40 | 127.1 | 125.0 | 123.8 | 23.0 | 80.5 | 5.2 | 3.7 | 3.0 |
| Niue | 2012 | 492 | 233 | 40 | 130.6 | 126.1 | 125.1 | 32.4 | 97.7 | 6.7 | 4.6 | NA |
| Qatar | 2012 | 1,057 | 422 | 36 | 118.2 | 116.5 | 115.4 | 29.3 | 94.0 | 5.1 | 4.2 | 3.7 |
| Republic of Moldova | 2013 | 2,095 | 784 | 42 | 133.2 | 130.3 | 128.5 | 26.4 | 86.0 | 5.1 | 4.5 | 6.1 |
| Rwanda | 2013 | 5,646 | 2,164 | 36 | 126.4 | 122.9 | 120.9 | 22.5 | 76.6 | 3.9 | 3.2 | 1.7 |
| Samoa | 2013 | 1,314 | 518 | 38 | 128.6 | 124.3 | 123.5 | 32.4 | 99.9 | 6.7 | 4.4 | 3.7 |
| Sao Tome and Principe | 2019 | 1,351 | 594 | 35 | 127.0 | 123.1 | 121.8 | 24.3 | 83.7 | 5.5 | 5.3 | 2.1 |
| Seychelles | 2004 | 849 | 396 | 42 | 128.0 | 124.9 | 122.7 | 26.4 | 88.2 | 5.7 | 5.4 | 3.2 |

*(Continued)*

**Table 1.** (Continued)

| Country | Year | Sample | Men | Age (years) | First SBP (mm Hg) | Second SBP (mm Hg) | Third SBP (mm Hg) | BMI (kg/m²) | Waist circumference (cm) | FPG (mmol/L) | TC (mmol/L) | 10-year CV risk |
|---|---|---|---|---|---|---|---|---|---|---|---|---|
| Solomon Islands | 2015 | 1,422 | 666 | 39 | 126.7 | 122.4 | 121.3 | 26.8 | 85.8 | 4.5 | 4.5 | 3.5 |
| Sri Lanka | 2015 | 3,181 | 1,294 | 42 | 127.3 | 125.1 | 123.9 | 23.0 | 82.4 | 4.7 | 4.1 | 3.1 |
| Sudan | 2016 | 5,692 | 2,110 | 37 | 131.9 | 127.8 | 126.3 | 23.3 | 83.6 | 4.7 | 3.9 | 4.2 |
| Tajikistan | 2017 | 1,924 | 855 | 38 | 134.3 | 130.4 | 128.2 | 25.5 | 82.0 | 5.2 | 3.9 | 2.8 |
| Timor-Leste | 2014 | 2,064 | 868 | 40 | 126.7 | 124.3 | 123.2 | 21.0 | 76.0 | 4.3 | 3.8 | 2.9 |
| Togo | 2011 | 1,186 | 546 | 36 | 128.0 | 124.3 | 123.3 | 23.2 | 80.7 | 4.3 | 4.4 | 1.7 |
| Tokelau | 2014 | 413 | 199 | 36 | 128.4 | 123.7 | 122.9 | 33.3 | 100.3 | 6.9 | 5.1 | NA |
| Turkmenistan | 2018 | 3,108 | 1,386 | 38 | 129.0 | 126.3 | 124.5 | 25.4 | 89.2 | 5.1 | 4.2 | 2.4 |
| Tuvalu | 2015 | 823 | 392 | 40 | 138.6 | 134.1 | 133.2 | 32.6 | 99.9 | 4.9 | 4.1 | NA |
| Uganda | 2014 | 2,976 | 1,287 | 35 | 129.3 | 124.8 | 122.7 | 22.7 | 78.5 | 4.0 | 3.5 | 1.6 |
| United Republic of Tanzania | 2012 | 1,555 | 717 | 42 | 136.4 | 132.8 | 130.6 | 22.8 | 83.1 | 4.8 | 4.5 | 3.1 |
| Uruguay | 2014 | 875 | 313 | 41 | 125.2 | 123.9 | 123.0 | 26.6 | 89.9 | 5.1 | 4.6 | 3.0 |
| Vanuatu | 2011 | 3,643 | 1,865 | 41 | 134.5 | 130.6 | 129.5 | 26.1 | 77.0 | 5.7 | 4.9 | 4.1 |
| Vietnam | 2015 | 2,517 | 1,076 | 42 | 122.6 | 119.3 | 118.0 | 22.0 | 76.6 | 4.0 | 4.5 | 2.8 |
| Zambia | 2017 | 2,791 | 1,135 | 35 | 127.0 | 124.2 | 122.3 | 23.0 | 79.1 | 5.0 | 3.4 | 1.9 |

Sample and men are absolute numbers. All other numeric variables are presented as means. Ten-year CV risk is 10-year predicted CV risk based on the 2019 World Health Organization Cardiovascular Risk Charts [16]. There are 6 countries with missing information for 10-year predicted CV risk because the risk model did not provide results for these countries (British Virgin Islands, Cook Islands, Niue, Nauru, Tokelau, and Tuvalu). The standard deviations for the numeric variables presented in this table are shown in S15 Table. BMI, body mass index; CV, cardiovascular; FPG, fasting plasma glucose; NA, not available; SBP, systolic blood pressure; TC, total cholesterol.

approaches. Conversely, using the second BP reading if the first BP measurement is ≥140/90 mm Hg does not appear to be a reasonable option because it yields large misdiagnosis rates.

## Over-diagnosis

Globally (S2 Table), as well as across the 6 regions (S3 Table), the simplified approach with the smallest over-diagnosis proportion was using the second BP reading if the first BP measurement was ≥140/90 mm Hg. Globally, this proportion was 3.03%; across regions this

**Table 2. Proportion (%) of missed hypertension cases for each of the 9 simplified approaches: Global results.**

| Simplified approach | Mean (%) | SD | Median (%) | Min (%) | Max (%) |
|---|---|---|---|---|---|
| 1st BP record | 10.18 | 3.45 | 10.08 | 2.21 | 18.88 |
| 2nd BP record | 5.82 | 2.41 | 5.56 | 1.21 | 12.50 |
| Average of 1st and 2nd BP records | 7.26 | 2.31 | 6.88 | 2.65 | 13.24 |
| 2nd BP record if 1st BP record ≥ 130/80 | 7.38 | 3.01 | 6.91 | 1.21 | 15.34 |
| 2nd BP record if 1st BP record ≥ 135/85 | 9.57 | 3.40 | 9.04 | 2.92 | 17.70 |
| 2nd BP record if 1st BP record ≥ 140/90 | 15.15 | 4.88 | 14.71 | 3.45 | 28.02 |
| 2nd BP record if 1st BP record = 130–145/80–95 | 5.62 | 2.44 | 5.35 | 1.21 | 13.86 |
| 2nd BP record if 1st BP record = 130–150/80–100 | 6.65 | 2.75 | 6.33 | 1.21 | 15.34 |
| 2nd BP record if 1st BP record = 130–155/80–105 | 7.07 | 2.91 | 6.73 | 1.21 | 15.34 |

All estimates shown in the tables (mean, SD, median, minimum, and maximum) are across the 60 countries included in the analysis. BP values given in millimetres of mercury. BP, blood pressure; max, maximum; min, minimum; SD, standard deviation.

**Table 3. Proportion (%) of missed hypertension cases for each of the 9 simplified approaches by WHO world region.**

| Region | Simplified approach | Mean (%) | SD | Median (%) | Min (%) | Max (%) |
|---|---|---|---|---|---|---|
| Africa | 1st BP record | 10.25 | 3.22 | 10.23 | 5.56 | 14.48 |
| | 2nd BP record | 5.45 | 1.84 | 5.48 | 2.88 | 8.68 |
| | Average of 1st and 2nd BP records | 7.34 | 2.14 | 7.21 | 3.85 | 10.34 |
| | 2nd BP record if 1st BP record ≥ 130/80 | 7.18 | 2.48 | 6.62 | 3.63 | 11.32 |
| | 2nd BP record if 1st BP record ≥ 135/85 | 9.51 | 3.19 | 9.75 | 4.70 | 14.66 |
| | 2nd BP record if 1st BP record ≥ 140/90 | 15.07 | 4.13 | 15.29 | 8.76 | 20.91 |
| | 2nd BP record if 1st BP record = 130–145/80–95 | 5.52 | 1.92 | 5.37 | 2.56 | 9.06 |
| | 2nd BP record if 1st BP record = 130–150/80–100 | 6.50 | 2.19 | 6.04 | 3.21 | 10.19 |
| | 2nd BP record if 1st BP record = 130–155/80–105 | 6.82 | 2.24 | 6.52 | 3.42 | 10.19 |
| Americas | 1st BP record | 14.90 | 2.82 | 14.11 | 12.50 | 18.88 |
| | 2nd BP record | 9.85 | 2.52 | 9.71 | 7.49 | 12.50 |
| | Average of 1st and 2nd BP records | 10.38 | 1.93 | 9.81 | 8.91 | 12.98 |
| | 2nd BP record if 1st BP record ≥ 130/80 | 11.77 | 3.54 | 11.60 | 8.56 | 15.34 |
| | 2nd BP record if 1st BP record ≥ 135/85 | 14.62 | 2.73 | 14.45 | 11.88 | 17.70 |
| | 2nd BP record if 1st BP record ≥ 140/90 | 23.32 | 3.61 | 22.99 | 19.25 | 28.02 |
| | 2nd BP record if 1st BP record = 130–145/80–95 | 9.25 | 3.87 | 9.10 | 4.95 | 13.86 |
| | 2nd BP record if 1st BP record = 130–150/80–100 | 11.39 | 3.97 | 11.15 | 7.92 | 15.34 |
| | 2nd BP record if 1st BP record = 130–155/80–105 | 11.64 | 3.71 | 11.60 | 8.02 | 15.34 |
| Eastern Mediterranean | 1st BP record | 8.91 | 4.12 | 9.31 | 2.21 | 15.22 |
| | 2nd BP record | 4.99 | 2.53 | 5.53 | 1.21 | 8.26 |
| | Average of 1st and 2nd BP records | 6.64 | 2.42 | 6.88 | 2.65 | 9.97 |
| | 2nd BP record if 1st BP record ≥ 130/80 | 6.49 | 3.22 | 7.10 | 1.21 | 10.88 |
| | 2nd BP record if 1st BP record ≥ 135/85 | 8.72 | 3.61 | 8.32 | 2.92 | 14.42 |
| | 2nd BP record if 1st BP record ≥ 140/90 | 13.37 | 6.63 | 13.99 | 3.45 | 23.91 |
| | 2nd BP record if 1st BP record = 130–145/80–95 | 5.12 | 2.22 | 5.51 | 1.21 | 7.25 |
| | 2nd BP record if 1st BP record = 130–150/80–100 | 5.94 | 2.84 | 6.26 | 1.21 | 9.42 |
| | 2nd BP record if 1st BP record = 130–155/80–105 | 6.30 | 3.04 | 6.75 | 1.21 | 9.96 |
| Europe | 1st BP record | 8.91 | 1.86 | 9.08 | 6.07 | 11.79 |
| | 2nd BP record | 3.83 | 0.60 | 3.96 | 3.03 | 4.62 |
| | Average of 1st and 2nd BP records | 5.59 | 1.30 | 5.90 | 3.56 | 7.51 |
| | 2nd BP record if 1st BP record ≥ 130/80 | 4.79 | 0.70 | 4.66 | 3.85 | 5.86 |
| | 2nd BP record if 1st BP record ≥ 135/85 | 6.59 | 0.91 | 6.52 | 5.36 | 7.81 |
| | 2nd BP record if 1st BP record ≥ 140/90 | 11.90 | 1.68 | 11.55 | 10.04 | 14.11 |
| | 2nd BP record if 1st BP record = 130–145/80–95 | 3.64 | 0.68 | 3.85 | 2.51 | 4.50 |
| | 2nd BP record if 1st BP record = 130–150/80–100 | 4.36 | 0.85 | 4.66 | 3.14 | 5.26 |
| | 2nd BP record if 1st BP record = 130–155/80–105 | 4.47 | 0.85 | 4.66 | 3.14 | 5.41 |
| Southeast Asia | 1st BP record | 8.89 | 2.55 | 8.38 | 5.07 | 12.12 |
| | 2nd BP record | 4.26 | 0.76 | 4.07 | 3.56 | 5.56 |
| | Average of 1st and 2nd BP records | 6.34 | 1.55 | 6.22 | 4.21 | 8.63 |
| | 2nd BP record if 1st BP record ≥ 130/80 | 5.11 | 1.09 | 4.64 | 4.12 | 6.81 |
| | 2nd BP record if 1st BP record ≥ 135/85 | 6.64 | 1.39 | 6.63 | 4.81 | 8.40 |
| | 2nd BP record if 1st BP record ≥ 140/90 | 12.14 | 3.06 | 11.03 | 8.42 | 16.31 |
| | 2nd BP record if 1st BP record = 130–145/80–95 | 4.25 | 1.10 | 4.04 | 3.01 | 5.90 |
| | 2nd BP record if 1st BP record = 130–150/80–100 | 4.83 | 1.16 | 4.43 | 3.61 | 6.70 |
| | 2nd BP record if 1st BP record = 130–155/80–105 | 5.01 | 1.10 | 4.50 | 4.12 | 6.81 |

(*Continued*)

**Table 3.** (Continued)

| Region | Simplified approach | Mean (%) | SD | Median (%) | Min (%) | Max (%) |
|---|---|---|---|---|---|---|
| Western Pacific | 1st BP record | 10.80 | 3.43 | 10.66 | 5.67 | 17.65 |
| | 2nd BP record | 7.29 | 1.96 | 7.38 | 3.62 | 10.76 |
| | Average of 1st and 2nd BP records | 7.89 | 2.38 | 7.12 | 4.18 | 13.24 |
| | 2nd BP record if 1st BP record $\geq$ 130/80 | 9.17 | 2.35 | 9.20 | 4.74 | 13.24 |
| | 2nd BP record if 1st BP record $\geq$ 135/85 | 11.41 | 2.41 | 11.67 | 7.80 | 16.91 |
| | 2nd BP record if 1st BP record $\geq$ 140/90 | 16.96 | 3.17 | 17.32 | 10.86 | 22.79 |
| | 2nd BP record if 1st BP record = 130–145/80–95 | 6.59 | 2.40 | 6.37 | 2.70 | 11.03 |
| | 2nd BP record if 1st BP record = 130–150/80–100 | 7.83 | 2.10 | 7.63 | 4.46 | 11.03 |
| | 2nd BP record if 1st BP record = 130–155/80–105 | 8.69 | 2.35 | 8.66 | 4.46 | 13.24 |

BP values given in millimetres of mercury. BP, blood pressure; max, maximum; min, minimum; SD, standard deviation; WHO, World Health Organization.

proportion varied between 1.92% (Americas) and 3.77% (Europe). The over-diagnosis proportions by region for the simplified approach based on the second BP reading only were around 5%: 4.48% (Americas), 5.05% (Southeast Asia), 5.65% (Western Pacific), 5.71% (Eastern Mediterranean), 6.11% (Africa), and 6.86% (Europe). The over-diagnosis proportions by region for the simplified approach based on the second BP reading if the first BP measurement was 130–

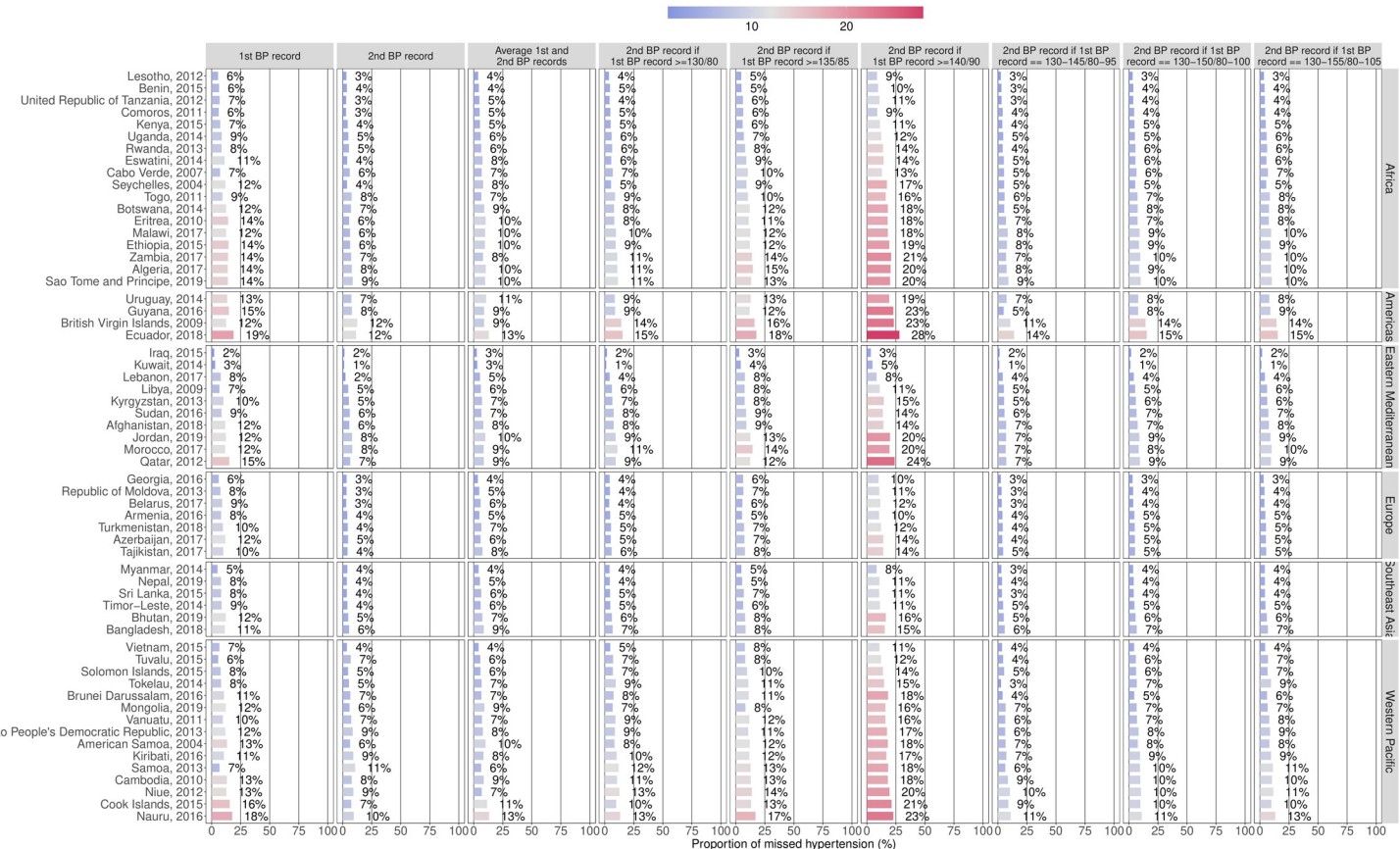

**Fig 1. Proportion (%) of missed hypertension cases for the 9 simplified approaches, stratified by country and region.** BP values given in millimetres of mercury. BP, blood pressure.

**Table 4. Multilevel regression models for missed hypertension diagnosis according to the 9 simplified BP screening approaches.**

| Simplified approach and independent variable | PR (95% CI) | *p*-Value |
|---|---|---|
| **1st BP record (*n* = 141,958)** | | |
| Female sex | 0.92 (0.85–0.99) | 0.028 |
| Age | 1.01 (1.00–1.01) | 0.001 |
| BMI | 1.02 (1.02–1.03) | <0.001 |
| Cardiovascular risk | 2.78 (1.03–7.46) | 0.043 |
| Random effects, country: region (variance) | 0.09 | |
| Random effects, region (variance) | 0.00 | |
| **2nd BP record (*n* = 141,958)** | | |
| Female sex | 0.88 (0.79–0.97) | 0.011 |
| Age | 1.01 (1.00–1.01) | 0.012 |
| BMI | 1.03 (1.02–1.04) | <0.001 |
| Cardiovascular risk | 2.51 (0.67–9.43) | 0.172 |
| Random effects, country: region (variance) | 0.10 | |
| Random effects, region (variance) | 0.00 | |
| **Average of 1st and 2nd BP records (*n* = 141,958)** | | |
| Female sex | 0.88 (0.80–0.96) | 0.003 |
| Age | 1.01 (1.00–1.01) | 0.001 |
| BMI | 1.03 (1.02–1.04) | <0.001 |
| Cardiovascular risk | 0.88 (0.25–3.09) | 0.841 |
| Random effects, country: region (variance) | 0.08 | |
| Random effects, region (variance) | 0.01 | |
| **2nd BP record if 1st BP record ≥ 130/80 (*n* = 141,958)** | | |
| Female sex | 0.91 (0.83–0.99) | 0.034 |
| Age | 1.00 (1.00–1.01) | 0.061 |
| BMI | 1.03 (1.02–1.04) | <0.001 |
| Cardiovascular risk | 3.20 (0.98–10.41) | 0.054 |
| Random effects, country: region (variance) | 0.11 | |
| Random effects, region (variance) | 0.00 | |
| **2nd BP record if 1st BP record ≥ 135/85 (*n* = 141,958)** | | |
| Female sex | 0.89 (0.82–0.97) | 0.006 |
| Age | 1.01 (1.00–1.01) | 0.019 |
| BMI | 1.03 (1.02–1.03) | <0.001 |
| Cardiovascular risk | 3.57 (1.29–9.88) | 0.0145 |
| Random effects, country: region (variance) | 0.09 | |
| Random effects, region (variance) | 0.00 | |
| **2nd BP record if 1st BP record ≥ 140/90 (*n* = 141,958)** | | |
| Female sex | 0.89 (0.84–0.95) | <0.001 |
| Age | 1.01 (1.01–1.01) | <0.001 |
| BMI | 1.03 (1.02–1.04) | <0.001 |
| Cardiovascular risk | 2.59 (1.14–5.84) | 0.022 |
| Random effects, country: region (variance) | 0.10 | |
| Random effects, region (variance) | 0.00 | |
| **2nd BP record if 1st BP record = 130–145/80–95 (*n* = 141,958)** | | |
| Female sex | 0.92 (0.83–1.02) | 0.123 |
| Age | 1.00 (1.00–1.01) | 0.208 |
| BMI | 1.03 (1.02–1.03) | <0.001 |

(*Continued*)

**Table 4.** (Continued)

| Simplified approach and independent variable | PR (95% CI) | p-Value |
|---|---|---|
| Cardiovascular risk | 2.42 (0.60–9.72) | 0.212 |
| Random effects, country: region (variance) | 0.09 | |
| Random effects, region (variance) | 0.00 | |
| **2nd BP record if 1st BP record = 130–150/80–100 (n = 141,958)** | | |
| Female sex | 0.89 (0.81–0.98) | 0.015 |
| Age | 1.00 (1.00–1.01) | 0.296 |
| BMI | 1.03 (1.02–1.04) | <0.001 |
| Cardiovascular risk | 2.69 (0.74–9.76) | 0.132 |
| Random effects, country: region (variance) | 0.09 | |
| Random effects, region (variance) | 0.00 | |
| **2nd BP record if 1st BP record = 130–155/80–105 (n = 141,958)** | | |
| Female sex | 0.90 (0.82–0.98) | 0.021 |
| Age | 1.00 (1.00–1.01) | 0.160 |
| BMI | 1.03 (1.02–1.04) | <0.001 |
| Cardiovascular risk | 2.59 (0.74–9.01) | 0.136 |
| Random effects, country: region (variance) | 0.09 | |
| Random effects, region (variance) | 0.00 | |

In all regression models the outcome was misdiagnosed (yes/no) for each simplified approach in comparison to the standard (average of the second and third BP measurements). The regression models included all independent variables together (i.e., only adjusted models were computed). BP values given in millimetres of mercury. 95% CI, 95% confidence interval; BP, blood pressure; BMI, body mass index; PR, prevalence ratio.

145/80–95 mm Hg were as low as 6.71% (Southeast Asia) and 6.79% (Americas), and as high as 9.61% (Europe) and 9.68% (Africa).

At the country level, the same pattern emerged (S1 Fig). The over-diagnosis proportion based on the second BP reading if the first BP measurement was ≥140/90 mm Hg was 1% in 7 countries. The over-diagnosis proportion based on the second BP reading alone was lowest in Kuwait and Cambodia (1%), whereas it was largest in Kyrgyzstan and Tajikistan (11%). The over-diagnosis proportion based on the second BP if the first BP measurement was 130–145/80–95 mm Hg was lowest in Kuwait (3%) and highest in Tajikistan (17%).

Even though the simplified approach of using the second BP record if the first BP measurement was ≥140/90 mm Hg yielded the lowest proportion of over-diagnosis, this approach had the largest proportion of misdiagnosis (as detailed in the previous section). The simplified approach based on the second BP reading alone had reasonable proportions of over-diagnosis and low rates of misdiagnosis (as detailed in the previous section).

## Cardiovascular risk profile amongst the misdiagnosed cases

Six countries (n = 3,216) did not have data for 10-year cardiovascular risk; thus, there were 54 countries (n = 141,958) included in this analysis. Descriptive statistics of the assessed risk factors per survey and stratified by the simplified approaches are available in S4 Table. The distribution of 10-year cardiovascular risk by country (hence survey), shown in S1 Fig, suggests there were no outliers.

Regarding the simplified approach based on the second BP measurement only, there were 34 countries where mean 10-year cardiovascular risk was not different between the missed and consistent hypertension groups (S3 Fig), yet the mean 10-year cardiovascular risk was slightly

lower in the former (mean = 4.17, SD = 1.78) than the latter (mean = 6.32, SD = 2.56) group. Moreover, higher 10-year cardiovascular risk was not associated with higher prevalence of being misdiagnosed (Table 4). The mean 10-year cardiovascular risk in the missed hypertension group was 21.64% of the mean in the consistent hypertension group in Azerbaijan, and 98.45% in Iraq (S14 Table). This suggests that in countries where there was no difference in the mean 10-year cardiovascular risk (e.g., Iraq; S3 Fig), the mean 10-year cardiovascular risk was very similar between the missed and consistent hypertension groups.

Regarding the simplified approach of using the second BP reading if the first BP measurement is 130–145/80–95 mm Hg, there were 34 countries where mean 10-year cardiovascular risk was not different between the missed and consistent hypertension groups (S8 Fig), yet the mean was slightly lower in the former (mean = 4.00; SD = 1.06) than the latter (mean = 6.13; SD = 2.40) group. Higher 10-year cardiovascular risk was not associated with higher prevalence of being misdiagnosed (Table 4). The mean 10-year cardiovascular risk in the missed hypertension group was 20.51% of the mean in the consistent hypertension group in Azerbaijan, and 94.17% in Iraq (S14 Table). This suggests that in countries where there was no difference in the mean 10-year cardiovascular risk (e.g., Iraq; S8 Fig), the mean cardiovascular risk was very similar between the consistent and missed hypertension groups (i.e., a ratio close to 94%).

These comparisons (including sampling sizes and p-values) for all of the simplified approaches are available in S2–S10 Figs and S5–S13 Tables.

The fact that mean cardiovascular risk was not significantly different between the missed and consistent hypertension groups for the 2 best-performing simplified approaches agrees with the results of the regression models, in which 10-year predicted risk was not associated with higher prevalence of misdiagnosis in these 2 simplified approaches. Together, these results may imply that in-country evaluations of these simplified approaches are needed to determine whether the missed cases have higher cardiovascular risk or not.

### Cardiovascular risk profile amongst the over-diagnosed cases

Regarding the simplified approach based on the second BP measurement only, there were 10 countries where mean 10-year cardiovascular risk was not different between the missed and consistent hypertension groups (S3 Fig), yet the mean 10-year cardiovascular risk was slightly lower in the former (mean = 5.43, SD = 2.20) than the latter (mean = 7.33, SD = 2.79) group. The mean 10-year cardiovascular risk in the missed group was 30.89% of the mean in the consistent hypertension group in Armenia, and 104.85% in Kuwait (S14 Table). This suggests that in countries where there was no difference in the mean 10-year cardiovascular risk (e.g., Kuwait; S3 Fig), the mean cardiovascular risk was similar between the missed and consistent hypertension groups.

Regarding the simplified approach of using the second BP reading if the first BP measurement is 130–145/80–95 mm Hg, there were 6 countries where mean 10-year cardiovascular risk was not different between the missed and consistent hypertension groups (S8 Fig), yet the mean was slightly smaller in the former (mean = 6.33; SD = 2.18) than the latter (mean = 8.17; SD = 3.25) group. The mean 10-year cardiovascular risk in the missed hypertension group was 35.41% of the mean in the consistent hypertension group in Zambia, and 99.87% in Kuwait (S14 Table). This suggests that in countries where there was no difference in the mean 10-year cardiovascular risk (e.g., Kuwait; S8 Fig), the mean cardiovascular risk was very similar between the consistent and missed hypertension groups (i.e., a ratio close to 100%).

### Potential correlates for missed hypertension cases

Age and BMI were positively associated with a higher probability of being misdiagnosed in all 9 simplified BP screening approaches, though with a small magnitude: The PR ranged between

1.00 and 1.03 (Table 4). Conversely, female sex (in comparison to male sex) was associated with a lower probability of being misdiagnosed with most of the 9 simplified approaches. Even though the associations for these independent variables were statistically significant, the strength of the associations may be negligible.

Ten-year cardiovascular risk showed a mixed profile, yielding strong positive associations for some of the simplified approaches. Overall, the simplified approaches in which 10-year cardiovascular risk was associated with higher prevalence of misdiagnosis could be less optimal. Conversely, simplified approaches in which 10-year cardiovascular risk did not show a strong association could warrant further attention, and be further considered.

The variability of the regression models was always larger between countries within regions, than between regions (consistent with Fig 1). This may imply that country-specific guidelines are needed for following simplified approaches, rather than 1 guideline for all countries in a region.

## Discussion

### Main findings

Leveraging 60 national surveys, we documented concordance between hypertension diagnosis based on the average of the last 2 of 3 BP measurements (standard approach) and 9 simplified approaches (e.g., second BP measurement if the first was above a threshold). The proportion of missed cases was lowest when using the second BP reading if the first BP measurement was 130–145/80–95 mm Hg, followed by using the second BP only. Notably, the former simplified approach would require a second BP measurement in some people only, reducing the total number of BP measurements, and therefore time and resources used, which could allow more people to be screened. We observed differences between countries within world regions. Also, we quantified the absolute cardiovascular risk in the missed hypertension group. In many countries, the mean cardiovascular risk was not different between the missed hypertension and consistent hypertension groups, yet the mean cardiovascular risk in the missed group was slightly lower than that in the consistent hypertension group.

Altogether, this research shows that simplified BP screening approaches may be sensible and could increase the number of people screened for hypertension, particularly in LMICs where screening for hypertension is still limited [4]. However, it would seem reasonable not to have a one-size-fits-all simplified approach. Although physicians may have reasonable concerns about missing hypertension cases with the simplified approaches, our findings suggest that missed cases may have slightly lower absolute cardiovascular risk than their peers with hypertension. Also, cardiovascular risk was positively associated with missed hypertension for only some simplified approaches. Future work with prospective cohorts should confirm this observation before simplified approaches are strongly recommended.

Across all countries, the proportion of missed hypertension in our study was similar to the proportion reported for simplified screening approaches in the US [14]: 10.2% versus 9.6% [14] for the first BP record, 5.8% versus 4.9% [14] for the second BP record, 7.3% versus 7.2% [14] for the average of the first and second BP records, and 7.4% versus 5.2% [14] for the second BP record if the first was ≥130/80 mm Hg. Conversely, our proportions of over-diagnosis were more than 2 times the proportions in the US: 14.4% versus 4.3% [14] for the first BP record, 5.8% versus 2.0% for the second BP record, 7.4% versus 2.0% for the average of the first 2 BP records, and 5.1% versus 2.0% [14] for the second BP record when the first was ≥130/80 [14]. The similar proportions for missed hypertension may suggest that the simplified BP screening approaches are sensible and little biased by measurement protocols. The higher over-diagnosis found in our study could be owing to different BP measurement protocols

between the STEPS surveys and the US national health survey [10,14]. Arguably, over-diagnosis would not be an unfavourable outcome, particularly when antihypertensive treatment is initiated at lower BP thresholds (provided the patient has other indications like history of cardiovascular disease or high cardiovascular risk) [7,9].

The results highlight that some of the 9 simplified approaches may lead to little misdiagnosis and are not associated with higher overall cardiovascular risk; however, we cautiously believe that further validation of these simplified approaches is warranted. Large prospective studies are needed to study the long-term cardiovascular outcomes for each simplified approach. Nevertheless, we would cautiously suggest considering 2 simplified approaches: (i) using only the second BP measurement and (ii) using the second BP reading if the first BP measurement is 130–145/80–95 mm Hg. If deemed necessary by local experts, these 2 simplified approaches could be implemented in screening programmes. Also, these 2 simplified approaches could be subject of further in-country validation analyses.

The WHO STEPS protocol [10], like other similar guidelines, recommends waiting 3 minutes between BP measurements. If there are 3 measurements (standard), and we assume that each measurement takes seconds, then measuring BP in 1 person could take at least 6 minutes. This would be equivalent to measuring BP in 10 people per hour. However, if a simplified approach is implemented, whereby, for example, only 2 measurements are required, the time invested to measure BP in 1 person would be approximately 3 minutes. In other words, we could measure BP in 20 people per hour, substantially increasing the number of individuals who could be screened for hypertension. The simplified approaches could save 50% of the time needed to measure BP in 1 person compared to current and standard guidelines.

Therefore, the potential applications of our work target several relevant scenarios. First, our work could influence May Measurement Month [12,13]. This is a global hypertension screening programme conducted yearly, and since 2016 it has covered more than 100 countries, benefiting over 1,000,000 people. This programme follows the 3-measurement protocol. Our work could inform future May Measurement Month campaigns by motivating discussion on whether fewer BP measurements could be taken, to maximise resources while reaching a much larger population. Second, our work could also influence future research and large health surveys. In addition to being used in the WHO STEPS surveys themselves, the WHO STEPS survey protocol has influenced other population-based health surveys worldwide, which would also take 3 BP measurements. Our work could spark interest in discussing whether 3 BP measurements are needed, or whether taking fewer measurements is a reasonable option to save resources that could be used to measure other relevant health variables. Third, in some clinics there may be a lack of sphygmomanometers or a shortage of personnel, limiting the number of people who can be screened for hypertension. Our work could deliver pragmatic approaches to optimise the protocols for BP measurement, to maximise the number of people who can be screened.

Our results showed large variability across countries within world regions. While simplified BP screening approaches may be a sensible and pragmatic alternative for screening large populations, a one-size-fits-all simplified approach may not be possible. Countries may need to find the optimal trade-off between the number of BP measurements and hypertension cases missed. Health organisations could set protocols for each country to define simplified BP screening approaches, so that these can be used in massive screening programmes [12,13].

Ten-year cardiovascular risk was positively associated with missed hypertension cases depending on the simplified approach used. Based on this, our results do not support relying on the first BP measurement only, for example. Conversely, our work may support using the second BP measurement or the average of the first 2 measurements, because 10-year cardiovascular risk was not associated with misdiagnosis amongst cases missed by these simplified approaches. The regression coefficients for absolute cardiovascular risk in some of the models,

depending on the simplified approach, did not reach statistical significance. We would argue that this signalled groups of missed cases in which cardiovascular risk was not (truly) associated with misdiagnosis. Given the large sample size included in the models, the results most likely show strong associations (or lack of association) than unstable results. Nevertheless, residual confounding could still be a possibility, even though we included the relevant and potential confounders that were available. Our results must be further validated with larger samples and, more importantly, with prospective cohort studies to examine the mid- and long-term cardiovascular outcomes of the missed and consistent hypertension groups [19].

## Strengths and limitations

Our study advances current knowledge on simplified BP screening approaches with estimates from 60 LMICs, and describes 10-year cardiovascular risk in the missed diagnosis, over-diagnosis, and consistent hypertension groups. Nonetheless, limitations should be acknowledged too. First, we analysed national health surveys with a standard protocol [10], which may not be equivalent to BP measurements in real life (e.g., massive screening programmes [12]). Unfortunately, massive screening programmes are not conducted routinely throughout the world, and where these occur, data are not available. Thus, population-based surveys are the only resource to expand the evidence about simplified BP screening approaches beyond a few countries. Second, because of data availability we only studied people aged 18–69 years. Recommendations derived from our work cannot be extrapolated to people ≥70 years of age. Third, this is a cross-sectional analysis. Although we documented that missed hypertension cases had slightly lower cardiovascular risk than those consistently diagnosed with hypertension, whether the missed cases went on to have worse cardiovascular outcomes than their peers who were consistently diagnosed with hypertension remains unknown. Large multi-country cohort studies are needed to strengthen this evidence [19]. Fourth, we used a standard cardiovascular risk score recommended for global use; however, this score may still have limitations. We used the cardiovascular risk score as a summary measure to characterise the overall cardiometabolic risk profile, not to make predictions about the cardiovascular outcomes in these populations. As discussed above, longitudinal studies are needed to characterise long-term cardiovascular outcomes for the simplified BP measurement approaches. Fifth, even though we pooled 60 national health surveys, all of which were nationally representative, for some countries the analysis included approximately 2,000 individuals after we applied our selection criteria. A sample size of approximately 2,000 people could be considered (rather) small, and could lead to variations in the estimates. However, we argue that these surveys were conducted using a standard and validated complex survey design in a random sample of the general population. Therefore, they provide informative results for the overall population in these countries. Given our selection criteria, the results for each country may not be representative of all people in the target population, but our results strongly characterise the patterns and profiles of the 9 simplified BP measurement strategies.

## Conclusions

Simplified BP screening approaches, to maximise resources and to reach much more people, appear to be sensible, with low rates of missed cases, amongst whom the absolute cardiovascular risk appears to be slightly lower than in the population with diagnosed hypertension. The fact that there was large variation in the percentage of missed hypertension cases for the different simplified approaches suggests that a one-size-fits-all approach should not be applied to all countries. More in-country research is needed to identify the factors affecting such variation among the countries.

## Supporting information

**S1 Analysis. Analysis code (R).**
(R)

**S1 Checklist. STROBE checklist.**
(DOC)

**S1 Cleaning. Code for data cleaning (R).**
(R)

**S1 Fig. Proportion over-diagnosed by simplified approach and country.**
(PDF)

**S2 Fig. Comparison of risk factors and 10-year absolute cardiovascular risk across classification status and by country based on the simplified approach first blood pressure only.**
(PDF)

**S3 Fig. Comparison of risk factors and 10-year absolute cardiovascular risk across classification status and by country based on the simplified approach second blood pressure only.**
(PDF)

**S4 Fig. Comparison of risk factors and 10-year absolute cardiovascular risk across classification status and by country based on the simplified approach average of the first and second blood pressure measurements.**
(PDF)

**S5 Fig. Comparison of risk factors and 10-year absolute cardiovascular risk across classification status and by country based on the simplified approach second blood pressure measurement if first is $\geq$130/80 mm Hg.**
(PDF)

**S6 Fig. Comparison of risk factors and 10-year absolute cardiovascular risk across classification status and by country based on the simplified approach second blood pressure measurement if first is $\geq$135/85 mm Hg.**
(PDF)

**S7 Fig. Comparison of risk factors and 10-year absolute cardiovascular risk across classification status and by country based on the simplified approach second blood pressure measurement if first is $\geq$140/90 mm Hg.**
(PDF)

**S8 Fig. Comparison of risk factors and 10-year absolute cardiovascular risk across classification status and by country based on the simplified approach second blood pressure measurement if first is 130–145/80–95 mm Hg.**
(PDF)

**S9 Fig. Comparison of risk factors and 10-year absolute cardiovascular risk across classification status and by country based on the simplified approach second blood pressure measurement if first is 130–150/80–100 mm Hg.**
(PDF)

**S10 Fig. Comparison of risk factors and 10-year absolute cardiovascular risk across classification status and by country based on the simplified approach second blood pressure measurement if first is 130–155/80–105 mm Hg.**
(PDF)

**S11 Fig. Histogram of 10-year absolute cardiovascular risk by country.**
(PDF)

**S1 Supplementary Flow Chart. Selection of study population.**
(DOCX)

**S1 Table. Countries and regions.**
(CSV)

**S2 Table. Proportion of over-diagnosis overall.**
(CSV)

**S3 Table. Proportion of over-diagnosis by region.**
(CSV)

**S4 Table. Description of cardiometabolic risk factors by country.**
(CSV)

**S5 Table. *p*-Values for S2 Fig.**
(CSV)

**S6 Table. *p*-Values for S3 Fig.**
(CSV)

**S7 Table. *p*-Values for S4 Fig.**
(CSV)

**S8 Table. *p*-Values for S5 Fig.**
(CSV)

**S9 Table. *p*-Values for S6 Fig.**
(CSV)

**S10 Table. *p*-Values for S7 Fig.**
(CSV)

**S11 Table. *p*-Values for S8 Fig.**
(CSV)

**S12 Table. *p*-Values for S9 Fig.**
(CSV)

**S13 Table. *p*-Values for S10 Fig.**
(CSV)

**S14 Table. Proportion of 10-year absolute cardiovascular risk comparing diagnosis status by country.**
(CSV)

**S15 Table. Standard deviations for Table 1.**
(CSV)

**S16 Table. Blood pressure measurement protocol by country.**
(XLSX)

**S1 WHO CVD Risk. Code for computing 10-year predicted absolute cardiovascular risk according to the 2019 World Health Organization Cardiovascular Risk Charts (Stata).**
(DO)

## Author Contributions

**Conceptualization:** Rodrigo M. Carrillo-Larco, Dinesh Neupane.

**Data curation:** Wilmer Cristobal Guzman-Vilca.

**Formal analysis:** Rodrigo M. Carrillo-Larco, Wilmer Cristobal Guzman-Vilca.

**Supervision:** Dinesh Neupane.

**Visualization:** Wilmer Cristobal Guzman-Vilca.

**Writing – original draft:** Rodrigo M. Carrillo-Larco.

**Writing – review & editing:** Rodrigo M. Carrillo-Larco, Wilmer Cristobal Guzman-Vilca, Dinesh Neupane.

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
