## [Editor Report · Decision Letter 0]

17 Nov 2021

Dear Dr Carrillo-Larco, 

Thank you for submitting your manuscript entitled "Simplified hypertension screening: misdiagnosis, over-diagnosis and cardio-metabolic characterisation in 60 low- and middle-income countries" for consideration by PLOS Medicine.

Your manuscript has now been evaluated by the PLOS Medicine editorial staff and I am writing to let you know that we would like to send your submission out for external peer review.

Please re-submit your manuscript within two working days, i.e. by Nov 19 2021 11:59PM.

Kind regards,

Beryne Odeny

PLOS Medicine

---

## [Decision Letter · Decision Letter 1]

25 Feb 2022

Dear Dr. Carrillo-Larco,

Thank you very much for submitting your manuscript "Simplified hypertension screening: misdiagnosis, over-diagnosis and cardio-metabolic characterisation in 60 low- and middle-income countries" (PMEDICINE-D-21-04705R1) for consideration at PLOS Medicine. 

Your paper was evaluated by a senior editor and discussed among all the editors here. It was also sent to independent reviewers, including a statistical reviewer. The reviews are appended at the bottom of this email and any accompanying reviewer attachments can be seen via the link below:

[LINK]

In light of these reviews, I am afraid that we will not be able to accept the manuscript for publication in the journal in its current form, but we would like to consider a revised version that addresses the reviewers' and editors' comments. Obviously we cannot make any decision about publication until we have seen the revised manuscript and your response, and we plan to seek re-review by one or more of the reviewers. 

We expect to receive your revised manuscript by Mar 18 2022 11:59PM. Please email us (plosmedicine@plos.org) if you have any questions or concerns.

We look forward to receiving your revised manuscript. 

Sincerely,

Beryne Odeny, 

PLOS Medicine

plosmedicine.org

1) Please include line numbers in your next draft

2) Please revise your title according to PLOS Medicine's style. Your title must be nondeclarative and not a question. It should begin with main concept if possible and include the setting of the study. Please place the study design (e.g., “a multi-country evaluation of surveys”) in the subtitle (i.e., after a colon).

3) Abstract:

a) Please structure your abstract using the PLOS Medicine headings (Background, Methods and Findings, Conclusions). Please merge Methods and Results into one section “Methods and Findings”

b) Please ensure that all numbers presented in the abstract are present and identical to numbers presented in the main manuscript text.

c) Please include the study design, regions and setting, total number of participants, month/year during which the surveys took place

d) Please include the specific important dependent variables that are adjusted for in the analyses.

e) In the last sentence of the Abstract’s Methods and Findings section, please describe the main limitation(s) of the study's methodology.

f) Abstract Conclusions: Please address the study implications without overreaching what can be concluded from the data.

5) Introduction: Please conclude the Introduction with a clear description of the study question or hypothesis.

6) Did your study have a prospective protocol or analysis plan? Please state this (either way) early in the Methods section. 

7) Please ensure that the study is reported according to the STROBE guideline, and include the completed STROBE checklist as Supporting Information. Please add the following statement, or similar, to the Methods: "This study is reported as per the Strengthening the Reporting of Observational Studies in Epidemiology (STROBE) guideline (S1 Checklist)." The STROBE guideline can be found here: http://www.equator-network.org/reporting-guidelines/strobe/

8) Please provide 95% CIs and p values, where relevant, for all estimates in the text and tables.

9) Please do not report p values to 3 decimal points 

10) In the Discussion section, please delete the titles “results in context” and “clinical medicine and public health implications” 

11) Please define the abbreviations in Tables and Figure e.g., BMI in Table 5.

12) Please remove the “Conflict of Interests”, “Data availability,” and “Funding” statements at the end of the main text. This information is captured in the metadata obtained in the submission form

13) References: 

a) Please select the PLOS Medicine reference style in your citation manager. In-text reference call outs should be presented as follows noting the absence of spaces within the square brackets, e.g., "... services [1,2]."

b) References should have no less and no more than six names before et al.

Comments from the reviewers:

Reviewer #1: This is a well-conducted study on simplified hypertension screening with regard to misdiagnosis, over-diagnosis and cardio-metabolic characterisation in 60 low- and middle-income countries. The study design, datasets, and statistical methods and analyses are mostly adequate. However, there are still a few issues needing attention, especially in presentation of the results.

1) The study is mostly descriptive and would be good to keep it descriptive to present all the facts on missing hypertentions in the 9 simplified screening approaches. The 10-year cardiovascular risk is just a risk score which itself has limitations in validation and accuracy, and they are a few others of similar kind. As all the surveys are cross-sectional studies, it would be good to focus on the observed facts rather than projections. Both Table 4 and Figure 2 are huge and not very informative and also on 10-year cardiovascular risk, suggest to move both to the supplementary information, and also tone down the claims on 10-year risks as subject to scrutiny and more work is needed.

2) Only around a couple of thousands paticipants in the survey in each country which is subject to variation and bias for the estimates. Can authors please go a bit comprehensive and critical in the limitation section to discuss the potential impact on the results of the study?

3) Table 5 on factors contributing to missed hypertension diagnosis. Age and BMI are mostly consistent but cadiovascular risk is not really, only 4 of 9 have significant results. Any interpretations and cautions to read the results?

Reviewer #2: This is a very well written, clear, and important paper.

The authors identified the share of misdiagnoses and over-diagnoses when using simplified hypertension screening approaches.

Given that hypertension screening needs to be increased globally in the fight against the epidemiological transition, more information on how screening efforts can be increased and made more efficient are highly relevant. 

The authors use adequate statistical methods in answering their research question.

I have only minor comments and suggestions - and the authors may choose to ignore them.

 - One sentence is a bit unclear (page 4 and 5): "We used the same predictors except for diabetes; the original model included history of diabetes whereas we included history and new diabetes cases (i.e., aware and unaware cases, the latter defined with fasting plasma glucose ≥126 mg/dL)"

 It should be made clearer which predictors this is referring to.

 - Perhaps the authors could add more details on how consequential simplified hypertension screenings may actually be for the CVD disease burden. This could be done by:

 - including references for HTN screening being still too low in LMICs (see this study https://www.thelancet.com/pdfs/journals/lancet/PIIS0140-6736(19)30955-9.pdf )

 - discussing how relevant it actually is to make screening time shorter. I wonder how much time is actually being saved by taking only e.g. only one measurement (could briefly mention wait time in between measurements). As the authors have pointed out, this may be particularly relevant in massive screening programs - but how frequent are these or should even be as screening efforts are being ramped up? 

Reviewer #3: This is an interesting approach to an important question. It's a big paper with a lot of analyses. I have one substantial criticism and a few smaller ones. 

The substantial criticism, which the authors responsibly describe in their limitations, is that the protocol-driven hypertension assessment of STEPS has poor alignment with real-world BP practices. A fair amount of evidence shows clinic values are less reliable, wider variability, and generally don't do the three-measure assessment this study is examining. It's testing a measurement approach that I basically think doesn't exist in the world. This makes the external utility of this work unclear to me. 

The smaller complaints all stem from, to my reading, them not quite succeeding at making a hard topic simple enough for the reader. There are a lot of analyses and I never quite found the "story" in their outcomes. There are 3 things that make this hard to write about. First is that diagnostic testing is a difficult field. The established terms of sensitivity and specificity are confusing and unnatural. Seemingly to avoid that they used non-standard terms like proportion missed and over-diagnosed. But these were hard to track and weren't reported consistently. The core findings - the classic 2x2 table of missed and overdiagnosed relative to true and non hypertensive - was clear in the abstract, but not elsewhere. 

Second, cross-national comparisons are hard to write because it's hard to see the story. A table of 60 countries is very hard to get a feel for or read, especially because it's unlikely that 2 countries of the same region actually have different phenomena. The manuscript dealt with this difficulty 3 different ways. They had a few tables with 60 countries, a few divided by continent, and some organized internationally with multilevel modeling (Table 5). I can see the urge for all of these, but it didn't help me understand what was going on. What do we learn from each? Do we think it's a national, regional, or single international phenomenon? 

This became especially difficult in the figures, which don't really work for me. They showed statistical significance, which is a feature of sample size, effect size, and random chance. In this case we know the effect is real - that the new measures do not precisely match the old - so there's no question we'd reach significance with enough sample. But I don't know the sample size of each of the 60 rows and this obscures the effect size, which is more interesting. I generally didn't find the story behind the figures that compelling compared to the general issue of learning how reliable the techniques are. (Though the fact, which appeared in a few ways, that the errors were larger in those with high risk and history of CVD is both concerning and interesting.)

The third difficulty was just that nine is a lot of comparisons. This made some tables feel overwhelming. But to address that, they didn't include all 9 at all spots in the paper, like the figures, even though we'd just seen them in a table. 

I think if this paper is resubmitted, I would recommend really deciding what the story is and telling that. Then, putting the details in the supplement and acknowledging in the paper that what is presented is post-hoc. I would put the missed and overdiagnosis in the same tables. I would report sensitivity and specificity somewhere. I would recommend grouping the countries in a way that feels meaningful and only putting all 60 in the main paper once. (Put more in an appendix.)

I have a few questions: 

1. Table 5 seems to show that higher risk people are are likely to have missed HTN. But Table 4 seems to show that missed hypertension have a risk of CVD that is roughly half the risk of those with consistent hypertension. How? 

2. In table 5 the HR of CV risk is ~3 in all examples except one where it's 0.89. Is that right? 

3. I would benefit by having them talk through the implications of the random effects findings in table 5.

4.Table 2 could use something like "by country" somewhere. I struggled to understand what the SD was the SD of, though I now see it's of the results of each country. 

5. I would probably merge tables 2 and 3, remove the min and max, and add columns for overdiagnosis into that table. 

Reviewer #4: Blood pressure measurement is key in the screening and management of high blood pressure. In the most comprehensive scenario patients would need multiple BP measurements

on separate occasions, and in the most pragmatic approach would require three BP measurements and taking the average of the last two which is standard approach. Three BP measurement may however be challenging in resource-constraint settings because of shortage health care workforce. Therefore, it will be necessary to have fewer BP measurements that would have values as close as possible to taking the average of the last two measurements which is currently the gold standard. Although attempts have been made to find simplified BP screening approaches, these simplified approaches were tested in three countries only and BP may vary between countries. The authors then decided to analyse national surveys in 60 low- and middle-income countries giving a large dataset. The authors further went ahead to relate the different measurement to their effect on cardiovascular risk score.

This paper is well written and a stimulation at looking for ways that can make blood pressure measurement especially during mass screening programmes to be more pragmatic.

Apart from country to country variations as stated by the authors is there any regional differences to the variation? Although one-size-fits will be applicable to different countries it will be nice as a more pragmatic approach for the authors based on their study to state the 2 or 3 methods out of the nine tested that they will recommend for countries to be tested.

Reviewer #5: Authors considered only BP measurement for hypertension diagnosis but currently having BP lowering drugs, earlier suggestion by a practitioner about hypertension as well as life style modification for hypertension management are part of hypertension definition which are ignored by authors. 

Reference 15 seems incorrect or broken.

[LINK]

---

## [Decision Letter · Decision Letter 2]

22 Mar 2022

Dear Dr. Carrillo-Larco,

Thank you very much for re-submitting your manuscript "Simplified hypertension screening: a multi-country evaluation of surveys" (PMEDICINE-D-21-04705R2) for review by PLOS Medicine.

I have discussed the paper with my colleagues and the academic editor and it was also seen again by two reviewers. I am pleased to say that provided the remaining editorial and production issues are dealt with we are planning to accept the paper for publication in the journal.

[LINK]

We look forward to receiving the revised manuscript by Mar 29 2022 11:59PM.   

Sincerely,

Beryne Odeny, 

PLOS Medicine

plosmedicine.org

Requests from Editors:

1. Please consider the suggested title, e.g., “Simplified hypertension screening methods across 60 countries: An observational study,” or similar.

2. Thank you for providing your STROBE checklist. Please replace the page numbers with paragraph numbers per section (e.g. "Methods, paragraph 1"), since the page numbers of the final published paper may be different from the page numbers in the current manuscript.

3. Please include p-values in tables where appropriate e.g., Table 4

4. Please replace the term “predictor” with “independent variable.” e.g., table 4

5. References – please include access dates for references with a weblink e.g. ref #6

Comments from Reviewers:

Reviewer #1: Many thanks authors for their great effort to improve the manuscript. The authors have addressed my comments very well. I am satisfied with the response and revision. No further issues needing attention.

Reviewer #5: Congratulations for great work

[LINK]

---

## [Editor Report · Decision Letter 3]

24 Mar 2022

Dear Dr Carrillo-Larco, 

On behalf of my colleagues and the Academic Editor, Dr. Joshua Z Willey, I am pleased to inform you that we have agreed to publish your manuscript "Simplified hypertension screening methods across 60 countries: An observational study" (PMEDICINE-D-21-04705R3) in PLOS Medicine.

PRESS

Sincerely, 

Beryne Odeny 

PLOS Medicine